# Cerebrospinal Fluid (CSF) Exchange with Artificial CSF Enriched with Mesenchymal Stem Cell Secretions Ameliorates Experimental Autoimmune Encephalomyelitis

**DOI:** 10.3390/ijms20071793

**Published:** 2019-04-11

**Authors:** Michael Valitsky, Sandrine Benhamron, Keren Nitzan, Dimitrios Karussis, Ezra Ella, Oded Abramsky, Ibrahim Kassis, Hanna Rosenmann

**Affiliations:** The Department of Neurology, The Agnes Ginges Center for Human Neurogenetics, Hadassah Hebrew University Medical Center, Jerusalem 91120, Israel; michaelvalitsky@gmail.com (M.V.); benhamron75@hotmail.com (S.B.); kfridel@gmail.com (K.N.); Dimitrios@hadassah.org.il (D.K.); ezrabest@yahoo.com (E.E.); ABRAMSKY@hadassah.org.il (O.A.); ibrahimk@ekmd.huji.ac.il (I.K.)

**Keywords:** CSF exchange therapy, mesenchymal stem cells, mesenchymal stem cell secretions, artificial CSF, experimental autoimmune encephalomyelitis

## Abstract

The complexity of central nervous system (CNS) degenerative/inflammatory diseases and the lack of substantially effective treatments point to the need for a broader therapeutic approach to target multiple components involved in the disease pathogenesis. We suggest a novel approach directed for the elimination of pathogenic agents from the CNS and, in parallel, its enrichment with an array of neuroprotective substances, using a “cerebrospinal fluid (CSF) exchange” procedure, in which endogenous (pathogenic) CSF is removed and replaced by artificial CSF (aCSF) enriched with secretions of human mesenchymal stem cells (MSCs). MSCs produce a variety of neuroprotective agents and have shown beneficial effects when cells are transplanted in animals and patients with CNS diseases. Our data show that MSCs grown in aCSF secrete neurotrophic factors, anti-inflammatory cytokines, and anti-oxidant agents; moreover, MSC-secretions-enriched-aCSF exerts neuroprotective and immunomodulatory effects in neuronal cell lines and spleen lymphocytes. Treatment of experimental-autoimmune-encephalomyelitis (EAE) mice with this enriched-aCSF using an intracerebroventricular (ICV) CSF exchange procedure (“CSF exchange therapy”) caused a significant delay in the onset of EAE and amelioration of the clinical symptoms, paralleled by a reduction in axonal damage and demyelination. These findings point to the therapeutic potential of the CSF exchange therapy using MSC-secretions-enriched-aCSF in inflammatory/degenerative diseases of the CNS.

## 1. Introduction

Degenerative diseases of the central nervous system (CNS) affect cognitive and motor functions and are the most common causes of chronic disability in adult life. Although impressive advances in understanding the mechanisms of these CNS diseases have been made, there is still no effective drug that substantially changes the progression of neuronal damage. Neuroinflammatory and neurodegenerative diseases have a complex and multi-factorial pathogenesis that involves several biological pathways, including aberrant protein interactions and aggregation, mitochondrial dysfunction, oxidative stress, inflammation, and apoptosis (reviewed in [1]).

It is now clear that targeting a single factor of the pathogenic cascade (e.g., a protein or a neurotransmitter) is insufficient for effective therapy, and it seems that targeting multiple disease related components could be more successful. Moreover, therapy may be more effective if directed to both eliminate the array of pathogenic agents from the CNS and enrich the CNS with an array of protective agents. In this way, the CNS milieu could change from a pathogenic to healthy one. A possible way to achieve such a change is by exchanging the fluid that bathes and is in direct contact with the whole CNS, i.e., the cerebrospinal fluid (CSF). The CSF provides a “sink action” by which the metabolic waste products formed in the nervous tissue during its metabolic activity diffuse rapidly into the CSF and are thus removed into the bloodstream as CSF is absorbed. It also provides a metabolic support and an active signaling milieu to the CNS, including the distribution of biologically active substances throughout the brain [2]. Changes in the composition of the CSF are evident in many neurodegenerative diseases, including disease-specific proteins, growth factors and neurotrophic agents [3,4,5]. Moreover, toxicity of the CSF from patients affected by neurodegenerative diseases has been reported [6,7]. Its immediate contact with the brain and its ability to swiftly distribute signals across vast distances in the CNS makes the CSF a suitable route for a therapeutic intervention aiming to change the CNS milieu.

We suggest here a novel approach for the treatment of degenerative/inflammatory diseases of the CNS, “CSF exchange therapy”. The principle of CSF exchange is similar to plasma exchange by plasmapheresis, which is in use for the treatment of autoimmune disorders [8]. Unlike plasma exchange, which can be done easily by removing the recipient’s plasma and replacing it with plasma from a healthy person, our proposed CSF exchange does not require the donation of CSF (that would demand the collection of high volumes of CSF by lumbar puncture in a donor) as a source for replacing the endogenous one. Our treatment protocol involves the use of artificial CSF (aCSF) supplemented with bioactive components to produce a biologically enriched aCSF. CSF exchange therapy involves two parallel stages: (1) removal of the endogenous (pathogenic) CSF and (2) its replacement with the biologically enriched artificial CSF, using an intracerebroventricular (ICV) CSF exchange procedure.

Simple transfer of CSF has been studied in animals, showing that infusion of CSF from healthy donor to recipient affected a wide range of behaviors, including sleep/wake and appetite (reviewed in [9]). In humans, it was reported that filtration of the CSF was beneficial in Guillain-Barré syndrome [10], but less effective in amyotrophic lateral sclerosis (ALS) patients [11]. CSF drainage was also tested in brain injury [12] and in CSF-fistula [13], with beneficial effects. An ad-hoc exchange of CSF was applied in one unique case aimed at removing accidental high levels of antibiotics that were given ICV and caused serial epileptic seizures in a patient with severe traumatic brain injury and occlusive hydrocephalus. The CSF was drained through a ventricular catheter, while mock CSF (ion solution with glucose) was infused into the lumbar subarachnoid space. The patient soon recovered from the epileptic seizures, with no adverse effects of the procedure [14]. With exception of this ad-hoc CSF exchange procedure using aCSF, to our knowledge, no other reports of CSF exchange therapy for chronic diseases of the CNS have been reported. Moreover, no studies using biologically enriched-aCSF for exchanging the CSF have been published.

We present here our results of CSF exchange therapy in the experimental autoimmune encephalomyelitis (EAE) mouse model of multiple sclerosis (MS), in which endogenous CSF was exchanged with aCSF enriched with secretions of human mesenchymal stem cells (MSCs). MSCs produce a variety of neurogenic, neuroprotective, and immunomodulatory agents [15,16,17,18,19,20,21], and have been shown to induce beneficial effects when transplanted in EAE-mice [22,23,24,25], stroke [26,27], traumatic brain injury [28], Parkinson’s disease [29], schizophrenia, and autism [30,31] as well as increased neurogenesis in adult mice [32]. MSC transplantation also showed indications of efficacy in clinical trials in MS and ALS patients [22,23,33,34,35]. We use here secretions of MSCs of human origin, having the advantage of exploring its potential for future therapeutic purposes. These cells are suitable for use in animals, since they present veto-like properties [36] and suppress host rejection [37].

The EAE-model served to establish the feasibility of CSF exchange therapy and its possible further application to other degenerative diseases of the CNS. A significant delay of EAE-onset with amelioration of clinical symptoms and reduced axonal damage and demyelination were noticed in the treated EAE-mice, pointing to the therapeutic potential of CSF exchange therapy with MSC secretions enriched-aCSF.

## 2. Results

### 2.1. Enriched-aCSF Increases Cell Viability of Neuronal Cell Lines

We used the secretions of the MSCs grown in aCSF (enriched-aCSF) for testing the effect of the secretions on the viability of neuronal cell lines (Study design in Appendix A).

PC12 were treated ON with enriched-aCSF containing secretions of 10 or 100 K/mL MSCs grown for 2 or 5 days in aCSF. While 2 day secretions of 10 or 100 K/mL MSCs in aCSF did not show an increase in PC12 cell viability (Figure 1a), the enriched-aCSF containing the secretions of MSCs growing for 5 days in aCSF showed a significant increase in PC12 cell viability relative to unenriched-aCSF treated cells. This increase was noticed in both 10 and 100 K/mL MSCs cell concentrations, showing a comparable effect (increase of 30.8% (*p* = 0.017) and 22.7% (*p* = 0.0065), respectively, relative to (unenriched) aCSF treated PC12 cells), indicating that a higher concentration at this range is not necessarily more effective (Figure 1b). We, therefore, preferred to use lower concentrations and further validated the beneficial effect of secretions of 10 and 25 K/mL in the growth of PC12 and SH-SY5Y cells. In both cell lines a significant increase in cell viability was noticed, with a dose effect in the SH-SY5Y cells (increase of 36.3% (*p* = 0.008), and 84.8% (*p* = 0.00013), respectively, relative to aCSF treated cells, with a stronger effect of the secretions of 25 K MSCs relative to 10 K (*p* = 0.0002) (Figure 1c)). In PC12 cell lines, the increase induced by both MSC concentrations was comparable (56.9% (*p* = 0.05), and 50.6% (trend *p* = 0.1) in Mann Whitney U test, respectively, relative to aCSF treated cells (Figure 1d)).

### 2.2. Enriched-aCSF Increases Cell Viability of Neuronal Cell Lines Exposed to Neurotoxins

We next tested whether the enriched-aCSF is also beneficial under a neurotoxic environment. Our results show that while Aβ reduced SH-SY5Y cell viability relative to control cells, enriched-aCSF significantly increased cell viability (10 K: increase of 81.9% (*p* = 0.0002), 25 K: increase of 83% (*p* = 0.00003) (Figure 2a)). A beneficial effect was also demonstrated under Aβ neurotoxicity in PC12 cells: while cell viability was reduced under Aβ, a significant increase in cell viability was noted in the enriched-aCSF treated cells (10 K: increase of 58.1% (Mann Whitney U, *p* = 0.02), 25 K: increase of 36.5% (*t*-test, *p* = 0.00004) (Figure 2b)). A similar trend of increased cell viability by enriched-aCSF treatment was noticed in SH-SY5Y cells exposed to H_2_O_2_, but without reaching a statistical significance (in Mann Whitney U test): (10 K: increase of 91.7%; 25 K: increase of 179.2%).

### 2.3. Enriched-aCSF Suppressed ConA Induced Splenocytes Proliferation

Proliferation in the presence of ConA was significantly suppressed in spleen lymphocytes treated with the enriched-aCSF compared to lymphocytes treated with (unenriched-) aCSF (an increase from 4200 ± 40.8 to 18,766 ± 1366.1 counts per minute (CPM) with ConA, and an increase from 6466 ± 131.2 to 46,683 ± 3566.2 CPM with ConA, in the enriched-aCSF vs. aCSF treated splenocytes, respectively, *p* = 0.0007). aCSF treated mice showed similar proliferation to splenocytes only (untreated) (Figure 3).

### 2.4. Enriched-aCSF Contains Neurotrophic Factors, Anti-Inflammatory Cytokines, and Anti-Oxidants

To characterize the composition of the MSC secretions in aCSF, “enriched-aCSF” (eCSF), we tested it for the presence of neuroprotective factors. We detected that the enriched-aCSF contained neurotrophic factors (brain-derived neurotrophic factor (BDNF) and ciliary neurotrophic factor (CNTF) and the anti-inflammatory cytokine transforming growth factor beta (TGF-β). We also detected anti-oxidants of two categories: enzymatic anti-oxidants and small molecules (Table 1).

Supernatants collected from MSCs grown in aCSF for 5 days (enriched-aCSF. eCSF) contain the neurotrophic factors brain-derived neurotrophic factor (BDNF) and ciliary neurotrophic factor (CNTF), the anti-inflammatory cytokine transforming growth factor beta (TGF-β), and anti-oxidant agents (enzymatic and non-enzymatic small molecules).

### 2.5. Short-Term MSC Secretions Enriched-aCSF (in/out) Exchange Therapy Induced a Short Term Amelioration of EAE Clinical Symptoms

We next used the MSC secretions enriched-aCSF (enriched-aCSF) to develop a CSF exchange procedure (Study design in Appendix A). This included insertion of the surgical cannulas into the ventricles and optimization of the rate and frequency of the exchange protocol to be tolerated by the animals. Feasibility of the CSF exchange procedure using the enriched-aCSF as a therapy for diseases of the CNS was then tested. As a disease model, we used the myelin oligodendrocyte glycoprotein (MOG)-EAE-mouse model for MS, an inflammatory disease with neurodegenerative components and clinical manifestations of motor deficits, which can be monitored daily. EAE-mice treated with “in/out” enriched-aCSF exchange therapy at days 6–8 following EAE-induction showed a trend towards a lower average clinical EAE-score at days 9–11 post EAE-induction (repeated ANOVA (f(1,16) = 3.73, *p* = 0.07), with a *t*-test for day 10 (4 days after starting treatment/2 days following end of treatment) revealing a significant difference in the average clinical score between treated and untreated mice (0 vs. 1.08 ± 0.30, respectively, *p* = 0.012). While the untreated mice developed the disease at day 9 following induction, the treated mice developed it only at day 11 (Figure 4a). When comparing the average cumulative score and the average maximal score at days 0–10, significantly lower scores were detected in the treated mice (*t*-test, *p* = 0.017, *p* = 0.012, respectively), with the average cumulative score showing a trend of being lower in the treated mice also at days 0–11 (i.e., until 3 days after end of therapy) (Mann Whitney U test, *p* = 0.06) (Figure 4b,c). However, from days 12–13 (i.e., 4–5 days after the end of therapy) the curves of the average score of treated and untreated mice merged, and no significant differences were noticed in the cumulative and maximal scores. Moreover, no significant difference was noticed at day 14 between the groups. Using the Kaplan Meier analysis for calculating the proportion of disease-free mice revealed significantly more EAE disease-free mice among the treated mice relative to the untreated ones (Log-Rank Test: *p* = 0.028), with the following median values for the 50% chance to stay free of disease symptoms: at day 11 for the treated and at day 8 for the untreated mice (Figure 4d). These results show a beneficial effect on disease onset and clinical symptoms already after 3 days of CSF exchange therapy using the MSC secretions enriched-aCSF. However, this effect lasted for only a few days.

### 2.6. Prolonged Amelioration of the EAE Clinical Symptoms during a Prolonged CSF Exchange Therapy

To determine whether the beneficial effect of therapy will be stronger and more prolonged, we next treated EAE-mice with a longer CSF exchange therapy, which was delivered during days 6–22 after disease induction (performed 5 times a week for 2 weeks). We used three protocols of prolonged CSF exchange therapy: “in/out” enriched-aCSF, “in” enriched-aCSF, and “in/out” aCSF (unenriched-aCSF). Since the clinical symptoms of the control EAE-mice in the various separate experiments, measured as average, cumulative, and maximal score, and the Kaplan Meier analysis were similar (no significant difference in one-way ANOVA analysis) we pooled the control EAE-mice results, and used them for comparison of the efficacy of the different CSF exchange therapies. Repeated ANOVA analysis showed a significant difference between the different treatment groups (f(2,24) = 9.59, *p* < 0.001)), with Tukey Post-hoc analysis showing significant differences between the control EAE-mice and each treatment group (*p* < 0.001, *p* = 0.035 and *p* = 0.001 for “in/out” enriched-aCSF, “in” enriched-aCSF, and “in/out” aCSF, respectively). While the “in/out” enriched-aCSF mice showed a significantly lower average score relative to controls at all days of follow-up of disease symptoms (*p* = 0.005, *p* = 0.013, *p* < 0.001, *p* = 0.001, *p* = 0.001, *p* = 0.005 for days 11, 12, 14, 15, 17, 18 post induction), the “in” enriched-aCSF mice showed a significantly lower average score relative to control-EAE (*p* < 0.001, *p* = 0.01, *p* = 0.002) only at days 11, 12, 14, respectively; the “in/out” aCSF mice showed a significantly lower score (*p* < 0.001, *p* = 0.01, *p* = 0.019) only at days 14, 15, 17 (with a similar trend (*p* = 0.066) also at day 18). A lower average score of “in/out” enriched-aCSF-mice was demonstrated relative to that of “in” enriched-aCSF-mice at days 15 and 17 (*p* = 0.06 and *p* = 0.03, respectively). These results show that the “in/out” enriched-aCSF therapy was the most effective one, affecting both time of onset (EAE-control mice develop the disease at day 10 while the treated mice at day 14 post induction) and disease progression. The “in” enriched-aCSF treatment, on the other hand, was only beneficial in delaying disease onset (EAE-control mice developed the disease at day 10 while the “in” enriched-aCSF-mice treated mice at day 12 post induction), and the “in/out” aCSF” treatment was most effective in interfering with disease progression rather than in delaying its onset (Figure 5a). A significant difference in the average cumulative score between the groups was noticed (one-way ANOVA (f(3,44) = 11.04, *p* < 0.001)); further analysis with Tukey Post-hoc showed that the (in/out) enriched-aCSF-mice and the (in/out) aCSF had lower scores compared to EAE-controls (*p* < 0.001, *p* = 0.008, respectively), and (in/out) enriched-aCSF-mice having lower scores than the (in) enriched-aCSF-mice (*p* = 0.004) (Figure 5b). A significant difference in average maximal score between the groups was noticed (one-way ANOVA (f(3,48) = 5.55, *p* = 0.003)); further analysis with Tukey Post-hoc showed that the (in/out) enriched-aCSF-mice had a lower score compared to control and (in) enriched-aCSF (*p* = 0.002, *p* = 0.02, respectively) (Figure 5c).

Kaplan Meier analysis revealed significant differences among the different groups in the proportion of disease-free mice (Log-Rank Test: *p* < 0.001), with each pair of groups showing significant differences (“in/out” enriched-aCSF-mice vs. all groups: EAE-control (*p* = 0.000387), “in” enriched-aCSF-mice (0.00507) and “in/out” aCSF-mice (0.0041); “in” enriched-aCSF-mice vs. EAE-control (*p* = 0.0415) and vs. “in/out” aCSF-mice (0.0141)), with the exception of the comparison between “in/out” aCSF-treated mice vs EAE-controls, which was not significant. The median value for the 50% chances to stay free of disease symptoms was 15, 13, 11, and 9 days for the “in/out” enriched-aCSF-mice, “in” enriched-aCSF-mice, “in/out” aCSF-mice and EAE-control, respectively (Figure 5d).

While the short-term treatment reached a decrease of about one degree of the average clinical score (effect at day 10: 1.08 ± 0.30 vs. 0 in control vs treated, respectively), the prolonged treatment induced a decrease of about two degrees of the average clinical score (at day 18: 2.75 ± 0.22 vs. 0.7 ± 0.2 in control vs. treated, respectively). The decrease in the average cumulative score was about 1.9 units in the short-term treatment (at days 0–11: 2.375 ± 0.68 vs. 0.416 ± 0.327 in control vs treated, respectively), and 11.2 degrees in the prolonged treatment (at days 0–18: 15.096 ± 0.077 vs. 3.8 ± 0.8 in control vs treated, respectively). The decrease in the average maximal score was approximately one degree in the short-term (at days 0–10: 1.08 ± 0.30 vs. 0 in control vs treated, respectively) and 1.8 units in the prolonged treatment (at days 0–18: 3.11 ± 0.21 vs. 1.3 ± 0.2 in control vs treated, respectively). As for median disease onset, the short-term therapy delayed the clinical onset by 3 days (50% free of disease at day 8 in control and at day 11 in treated mice), and by 6 days in the prolonged therapy (50% free of clinical disease at day 9 in control and at day 15 in treated animals). This indicates a stronger and more prolonged beneficial effect of the prolonged CSF exchange therapy.

### 2.7. CSF Exchange Therapy Reduced Axonal Damage and Demyelination

We performed Bielschowsky silver impregnation for axonal pathology in order to evaluate the effect of the CSF exchange therapy on the degree of axonal damage in the cortex of EAE mice. Significantly lower axonal damage was observed in the “in/out” enriched-aCSF treated mice relative to the EAE-control mice (1.88 ± 0.46 (*n* = 7) vs. 2.57 ± 0.22 (*n* = 11), respectively, *p* = 0.048)). A similar trend of reduced axonal damage was also detected in the “in” enriched-aCSF-mice relative to the EAE-control mice (2.06 ± 0.22 (*n* = 9) vs. 2.61 ± 0.23 (*n* = 7), respectively, *p* = 0.09) while there was no difference between the “in/out” aCSF-treatment and the EAE-control (2.58 ± 0.20 (*n* = 8) vs. 2.28 ± 0.32 (*n* = 10), respectively, *p* = 0.203) (Figure 6a).

LFB staining for the degree of demyelination following CSF exchange therapy revealed a trend of less demyelination in the “in/out” enriched-aCSF treated-mice relatively to the EAE-control mice in the cortex (2.58 ± 0.22 vs. 2.98 ± 0.14, respectively, *p* = 0.09). Moreover, a significantly lower degree of demyelination was noticed in the “in” enriched-aCSF-mice relative to the EAE-control mice (2.69 ± 0.19 vs. 3.21 ± 0.09, respectively, *p* = 0.02), but there was no difference in demyelination between the “in/out” aCSF-treatment and the EAE-control (2.56 ± 0.187 vs. 2.70 ± 0.255, respectively, *p* = 0.314) (Figure 6b).

However, there was no difference in the number of perivascular cell infiltrates, entering from the periphery, between the treated mice and the EAE-controls (in the “in/out” enriched-aCSF treated vs EAE-control: no. of infiltrates and cells/infiltrate: 9.33 ± 1.48 vs. 6.08 ± 1.1 (*p* = 0.102), and 26.00 ± 3.42 vs. 21.04 ± 1.96 (*p* = 0.195), respectively. In the “in” enriched-aCSF treated vs EAE-controls: no. of infiltrates and cells/infiltrate: 3.90 ± 1.09 vs. 2.79 ± 1.24 (*p* = 0.512), and 18.06 ± 1.45 vs. 21.39 ± 2.31 (*p* = 0.255), respectively. In the “in/out” aCSF treated vs EAE-control: no. of infiltrates and cells/infiltrate: 5.59 ± 1.09 vs. 3.53 ± 0.91 (*p* = 0.14), and 20.47 ± 3.28 vs. 21.16 ± 2.10 (Mann Whitney U test, *p* = 0.993), respectively).

## 3. Discussion

In this study we present CSF exchange therapy, a novel therapeutic approach aimed at exchanging the endogenous pathogenic CSF with a new and healthy one, through drainage of the endogenous CSF and its continuous replacement with aCSF enriched with secretions of the MSCs. Our daily CSF exchange procedure at a rate of 5 μL/h exchange for 3 h can affect about 40% of the 35 μL circulating CSF in the mice [38]. We demonstrate that MSCs grown in aCSF secrete nourishing/neurotrophic, anti-inflammatory, and antioxidant factors. The MSC secretions enriched-aCSF induced an increase in the cell viability of neuronal cell lines and protection from neurotoxins, as well as suppressed splenocyte proliferation. In vivo CSF exchange therapy in EAE-mice using MSC secretions-enriched-aCSF induced a significant delay in disease onset and amelioration of the clinical symptoms, as well as reduction in axonal damage and demyelination.

While the rationale of the CSF exchange therapy approach, which lies in its multi targeting mechanism (its elimination of toxic components with parallel enrichment with a wide array of protective agents), may be relevant for various diseases of the CNS, we first tested and optimized the protocol in the EAE model for MS, an inflammatory and degenerative disease of the CNS. Our results show that elimination of endogenous CSF, and its replacement with MSC secretions enriched-aCSF ((in/out)-enriched-aCSF), delayed EAE onset and reduced the clinical score with indications of reduced axonal damage and demyelination. The clinical effect (onset and clinical score) was stronger and more persistent in the prolonged CSF exchange therapy than in the short-term treatment, with the effect still evident a few days after treatment.

Comparison of the effect of the prolonged (in/out)-enriched-aCSF to that of CSF exchange with unenriched-aCSF ((in/out)-aCSF), or infusion with enriched-aCSF without the elimination of the endogenous one ((in)-enriched-aCSF) (all of them in the prolonged treatment protocol), indicated that these two latter treatments also could have a beneficial effect on the clinical manifestations of EAE. However, while the (in/out)-enriched-aCSF treatment was effective in both delaying the disease onset and reducing the clinical score, this was not the case with the other two treatment protocols tested. The (in)-enriched-aCSF treatment delayed only the disease onset but affected less the clinical score, and the (in/out)-aCSF did not affect the onset but affected the clinical score at later stages of the disease. Overall, these effects were relatively less robust than using the (in/out)-enriched-aCSF procedure. The beneficial effect of the (in)-enriched-aCSF treatment (infusion only) in delaying the onset can be attributed to the protective molecules present in the enriched-aCSF and their immunomodulation properties, thereby interfering with the lymphocyte pro-inflammatory potency in the CNS, while the (unenriched) aCSF (even accompanied by elimination of endogenous CSF) lacks part of these mechanisms of action and, therefore, does not interfere with the immune mediated process taking place at the early stages of EAE. According to our data, it seems that the (in/out)-aCSF is beneficial at a later stage of the disease, when toxic elements are produced as a result of the inflammatory demyelination and axonal damage; elimination of such products from the CNS by CSF drainage may provide an effective way for slowing the progression of neuronal damage. This may explain the lower efficacy of (in)-enriched-aCSF treatment at later stages of EAE, as no elimination of endogenous CSF is performed. However, some additional indirect beneficial effects of the CSF exchange may take place, to a certain degree, through increased self-drainage of the endogenous CSF (particularly in the (in)-enriched-aCSF protocol), or increased self-production of fresh endogenous CSF (in the (in/out)-aCSF, which lacks the infusion of beneficial anti-inflammatory and neurotrophic components). In any case, amelioration of all stages of the EAE disease tested here, from onset to later stages, are induced by the (in/out)-enriched-aCSF therapy. The fact that MSCs have immunomodulatory properties has been reported [15,16,17,18,19,20,21], and this may explain our findings of the in vitro and in vivo modulation by the enriched-aCSF. We did not detect a decrease in infiltrate burden in the treated mice; such an effect can possibly be detected if tested at earlier stages, when the immune activation cascade starts. The enriched-aCSF had a pronounced effect on in vitro lymphocyte proliferation, pointing to the enhanced immunomodulatory effect of the enriched-aCSF. This is further supported by our finding that MSCs secrete anti-inflammatory cytokines when growing in aCSF. Testing CSF exchange with MSC secretions-enriched-aCSF in a non-inflammatory model of demyelination may shed light on the immunomodulatory role of this effect. A previous study failed to show protection by MSCs in a non-inflammatory model of neurological disease [39]. However, another study, in vitro and in vivo, has shown that MSCs can promote survival and axonal myelination in sensory dorsal root ganglia neurons and may be effective in non-inflammatory models of demyelination [40]. It is, therefore, possible that the effect of enriched-aCSF is mediated not only via immunomodulation, but also through direct effects on survival and axonal myelination. In fact, the presence of neurotrophic factors, anti-inflammatory-cytokines, and anti-oxidants in the enriched-aCSF may support the broad beneficial effect, as these agents have been reported to ameliorate EAE-symptoms, reducing inflammation as well as demyelination and axonal damage [41,42,43,44,45,46]. The possibility of a double mechanism of action may be of major clinical relevance, for possible application of CSF exchange protocols in relapsing remitting and progressive MS, but also in other neuroimmune and neurodegenerative diseases.

MSCs transplantation alone has been applied and tested with strong indications of beneficial effects in animal models of MS [22,23,24,25], stroke [26,27], traumatic brain injury [28], PD [29], schizophrenia, and autism [30,31], with encouraging indications in pivotal clinical trials in MS and ALS patients [22,23,33,34,35]. Many of the beneficial effects of MSC have been shown to be mediated by humoral mechanisms and the secretion of neurotrophic and immunomodulatory factors [15,16,17,18,19,20,21]. Here we used, for the first time, only the secretions of MSCs to produce an artificial enriched-CSF for ICV infusion, and in parallel, we exchanged the pathogenic CSF to maximize the beneficial effects. The concept of the induction of direct beneficial effects on the CNS by CSF exchange therapy (delivery of the enriched-aCSF into the ventricles) seems to be in accord with our previous studies, showing that local (intraventricular) transplantation of MSCs is more effective than intravenous administration [22].

Our approach of CSF exchange using MSC secretions may have the advantage of allowing repeated exchange treatments using frozen enriched-aCSF without the need of repeated growing of the cells and even without the necessity of autologous origin, since molecules, but not cells, are being transferred. The feasibility of CSF exchange therapy reported here in the EAE model might possibly be applied to other neurodegenerative diseases of the CNS. Just as importantly, while most therapeutic modalities for neurodegenerative diseases are preventive treatments, our approach of CSF exchange therapy could be beneficial for fully developed neurological diseases through a repeated application of the proposed CSF exchange protocol. Additional studies are warranted to provide further support to our findings, including studies in primary neuronal cells, as well as animal studies for testing the efficacy of the treatment once applied after the development of first clinical symptoms.

## 4. Materials and Methods

### 4.1. Cell Culture Studies

#### 4.1.1. Preparation of MSCs were Performed According to Our Reported Protocol

##### Bone Marrow (BM) Aspiration

Fresh BM, which was the source of MSCs, was aspirated from MS patients for cell transplantation as part of the clinical trials conducted in The Multiple Sclerosis Center, Department of Neurology, Hadassah (under the Helsinki Ethics committee approval, registered at NIH, NCT02166021), using autologous transplantation therapy, based on the previous results showing that MSCs derived from EAE-mice suppress EAE and have similar biological properties to those of the MSCs from healthy donors [22,23]. BM aspiration was performed according to the routine procedure in the Hadassah medical center from the patient’s iliac-crest under local anesthesia and sedation by an anesthetist. BM (~100 mL) was aspirated using aspiration needles into heparin containing sterile bags (Macopharma, Tourcoing, France).

##### Separation of Mononuclear Cells (MNCs) from the Whole BM

BM aspirates were transferred from the heparin containing BM aspiration bags into sterile 50 mL conical tubes (Corning, NY, USA) using two spike tubing sets (Macopharma) and diluted 1:1 (*v*:*v*) in Hank’s Balanced Salt Solution (HBSS, Sigma-Aldrich, Rehovot, Israel), and MNCs were separated from the total BM cells by the Ficoll density gradient (1.073 gr/mL) centrifugation (GE Healthcare, Uppsala, Sweden). Diluted BM aspirates were transferred in barrier-containing 50 mL tubes (LEUCOSEP™, Greiner-bio one, Frickenhausen, Germany) prefilled with 15 mL Ficoll and centrifuged for 10 min, 1000× *g*, 24 °C. The MNC layer was removed under sterile conditions, transferred into 50 mL sterile tubes, and diluted with 30 mL CTS™ DPBS (Gibco, Paisley, UK). Cells were centrifuged twice for 10 min, 1000 rpm, 24 °C. Cells were re-seeded in “complete culture media” containing Nutristem™ XF Basal Media supplemented with Nutristem™ (Biological Industries, Beit Haemek, Israel) Supplement media for further processing.

##### Propagation of MSCs

MNCs were counted and cell viability was evaluated using trypan-blue dye staining (Sigma-Aldrich). MNCs were washed and re-suspended with Nutristem XF™ complete media and seeded on Cell-Stack™ (636 cm^2^, Corning) pre-coated with Attachment Solution XF ™. The cells were incubated in a 37 °C/5% CO_2_ humidified incubator for 48 h at a seeding density of 100 K cells/cm^2^. Under microscopy, non-adherent mononuclear cells were seen floating in the culture supernatant and plastic-adherent MSCs attached to the flask surface. The culture supernatant containing the non-adherent mononuclear cells was removed, and the adherent cells were gently washed with 100 mL DPBS. The step from MSCs seeding to harvesting was named Passage 0 (P0). P0 cells were incubated in a 37 °C/5% CO_2_ humidified incubator and the growth medium was replaced twice a week with fresh complete NutriStem™ XF growth medium until 80–90% confluency, but for no more than 12 days. Each subculture cycle was counted as a new passage. The cultures were sub-cultivated up to Passage 3. For sub-culturing MSCs, the culture supernatant was removed from the flask and a CTS™ TrypLE™ Select solution was added to each flask. The flask was incubated for 8 min at 37 °C and culture medium was added to each CellStack™ to inactivate the enzymatic action. The detached cell suspensions were transferred into centrifuge tubes, washed, re-suspended in growth medium, counted and reseeded at a density of 5 K cells/cm^2^ in 120 mL of NutriStem™ XF growth medium in new CellStack™. The cultures were then incubated in a 37 °C/5% CO_2_ humidified incubator.

##### Characterization of Isolated Human MSCs

Isolated MSCs were characterized by flow cytometry analysis (FACS) of surface antigen expression, according to The International Society for Cellular Therapy (ISCT) recommendations. MSCs of MS patients were tested for the expression of the surface markers CD105, CD73 and CD90 (>95% positive) and for lack of expression (<2% positive) of CD34, CD45, CD79alpha, CD19, and HLA-DR. The stained cells were counted and analyzed with CYTOMICS FC500 (Beckman Coulter, Munich, Germany).

#### 4.1.2. Preparation of aCSF Enriched with Secretions of MSCs

MSCs were grown in aCSF and the supernatant was collected and frozen at −20 °C to be used as MSC secretions-enriched aCSF (“enriched-aCSF”) for in vitro studies (treatment of neuronal cell lines and spleen lymphocytes) and in vivo in EAE-mice (Study Design presented in Appendix A).

#### 4.1.3. Testing the Viability of Neuronal Cell Lines Treated with the Enriched-aCSF

To find the optimal conditions (MSCs concentration and growing time in aCSF) for achieving effective enriched-aCSF, 10–100 K/mL MSCs were grown in aCSF for 2 and 5 days, and the enriched-aCSF was collected and tested for its effect on the viability of pheochromocytoma cells (PC12, provided by ATCC) following overnight (ON) treatment. This was followed by using enriched-aCSF from 10–25 K/mL MSCs growing in aCSF for 5 days, for testing its effect on the viability of PC12 as well as of neuroblastoma cells (SH-SY5Y, provided by ATCC). The effect of enriched aCSF on the viability of the neuronal cell lines was also tested under exposure to neurotoxins by treating the cells ON with the enriched-aCSF, followed by addition of 0.5 mM H_2_O_2_ (Sigma) for 4 h, or the addition of 4 μM Aβ25-35 (Sigma) (pre-aggregated at 37 °C for 10 min) for ON. Viability was determined, at least in 3–4 replicates per group, by MTT assay [47].

#### 4.1.4. Measurement of Secreted Neurotrophic Factors, Anti-inflammatory Cytokines, and Anti-oxidant Capacity in the Enriched-aCSF

MSCs (600 K/mL) were grown in the aCSF in 24-well plates (Nunc, Roskilde, Denmark) for 5 days. Supernatants (enriched-aCSF) were collected, and frozen until used for testing the levels of the neurotrophic factors—brain-derived neurotrophic factor (BDNF) and ciliary neurotrophic factor (CNTF)—and for the levels of the anti-inflammatory cytokine: transforming growth factor beta (TGF-β) (ELISA kits, R&D Systems, Minneapolis, MN, USA). The total anti-oxidant capacity, including both enzymatic (catalase and peroxidase) and non-enzymatic capacity (small molecules, ascorbic acid, glutathione, carotenes, α-tocopherol, and ubiquinol), was measured. When adding a protein-masking reagent, only the anti-oxidant activity of the small molecules was determined (Sigma kit).

#### 4.1.5. In Vitro Proliferation of Spleen Lymphocytes (Splenocytes) Treated with the Enriched-aCSF

Splenocytes were excised from C57BL mice and cultured as single-cell suspensions. Splenocyte proliferation was assayed in vitro by ^3^H-thymidine incorporation. All cultures were carried out in triplicate in 96-well, flat-bottom, microtiter plates. The assay was carried out by seeding 400 K cells/well in 0.2 mL RPMI medium (Sigma) supplemented with 2.5% FCS, 1 mM l-glutamine, and antibiotics. Basal ^3^H-thymidine incorporation was determined, and in response to Concanavalin A (ConA, 1 µg/mL). The cultures (splenocytes only, splenocytes treated with aCSF or with the enriched-aCSF (diluted with 1:1 with the splenocyte medium)) were incubated for 48 h in a 37 °C/5% CO_2_ humidified atmosphere and pulsed for 16 h with ^3^H-thymidine (1 µCi/well). Cells were harvested on fiberglass filters using a multi-harvester and the radioactivity was counted (CPM) [23].

### 4.2. Animal Studies

The experiments were performed in accordance with local and international regulations and were approved by the Hebrew University of Jerusalem University Ethics Committee (Identification code: MD-13-13763-4; Date of approval: 16 October 2013).

#### 4.2.1. Insertion of the CSF Exchange Device

Nine-week-old female C57BL6 mice went through a standard stereotaxic surgical procedure for the insertion of the CSF exchange device, as follows: the animals were anesthetized (ketamine 3 mg/kg, cepetore 0.03 mg/kg, i.p.) and placed in a Kopf stereotaxic frame, and an incision was made to expose the skull. Small holes were drilled above the lateral ventricles, and the 2.5 mm guide cannula (RWD-900-0062-060) was cemented with dental acrylic cement mixed with cyanoacrylate glue to secure the cannula to the skull. Cannula dummies (RWD-900-0062-131 2) were inserted into the guide cannulas to keep them patent. Stereotaxic coordinates used for the ICV injection in mm from the bregma: −0.2 mm anteroposterior, ±1 mm mediolateral, −2.3 mm dorsoventral.

#### 4.2.2. Induction of EAE

One week after insertion of the device, chronic EAE was induced by immunization of the mice with an emulsion (0.2 mL) containing 300 µg purified myelin oligodendrocyte glycoprotein (MOG) 35–55 peptide in phosphate-buffered saline (PBS) and an equal volume of complete Freund’s adjuvant containing 5 mg of H37Ra (Difco Laboratories, Detroit, MI, USA). In addition, Bordetella pertussis toxin (300 ng in 0.2 mL PBS) was injected intraperitoneally the same day and 48 h later. Animals with EAE were scored daily for neurological symptoms according to the EAE clinical severity scale: 0 = asymptomatic; 1 = partial loss of tail tonicity; 2 = tail paralysis; 3 = hind limb weakness; 4 = hind limb paralysis; 5 = 4-limb paralysis; 6 = death [48].

#### 4.2.3. CSF Exchange Therapy

For CSF exchange therapy, mice were moved to individual therapy cages. The dummy cannula was removed and replaced by the injector cannula secured with fixing screw (Fixing Screw-Connecting Double Injector and Guide RWD-900-0062-521). In order to minimize stress and enable free spatial movement, all infusion (“in”) and withdrawal (“out”) tubing were connected to a low-torque dual channel swivel (Instech, 375/D/22LT), mount to counter-balanced lever arm (Instech, MCLA).

At day 6 after EAE induction, the CSF exchange therapy was started. In the first part of the study EAE-mice (*n* = 6) went through a short-term CSF exchange (“in/out”) therapy of 3 days (days 6–8 following EAE induction), as follows: infusion of MSC secretions enriched-aCSF into one ventricle (“in”), and withdrawal of endogenous CSF from the other ventricle (“out”), at a rate of 5 μL/h for 3 h. As control we used the EAE-mice to which the CSF exchange device was inserted but did not go through the CSF exchange therapy (*n* = 12). This experiment (#1) served us as a proof of concept, and for establishing a treatment protocol that was tolerated by the EAE-mice. This experiment was followed by additional experiments in which the treatment was more prolonged (performed at days 6–18 following disease induction, 5 times a week, at a rate of 5 μL/h for 3 h). Altogether, 3 additional prolonged experiments with different CSF therapy protocols were performed, as follows: In experiment #2 EAE-mice received “in/out” CSF exchange therapy with MSC secretions enriched-aCSF (*n* = 5) and compared to non-treated EAE-controls (*n* = 6). In experiment #3, EAE-mice received only infusion of MSC secretions enriched-aCSF into one ventricle (“in”) without withdrawal of endogenous CSF (*n* = 9), vs non-treated EAE-controls (*n* = 7). In experiment #4, “in/out” CSF exchange therapy was performed with aCSF only (non- enriched) (*n* = 8) vs. non-treated EAE-controls (*n* = 13).

#### 4.2.4. Histological Examination

Mice were anesthetized with a lethal dose of pentobarbital and perfused via the ascending aorta with 4% paraformaldehyde, and brains were removed and preserved at −80 °C. Serial 10 μM sections were made. The modified Bielschowsky staining was used to evaluate the axonal loss, using the following score: 0, normal axonal density; 1, focused mild to moderate axonal loss; 2, scattered mild to moderate axonal loss; 3, focused severe axonal loss; and 4, scattered severe axonal loss. Luxol fast blue staining was used to grade the demyelination, as follows: 0, no demyelination; 1, a few scattered naked axons; 2, small groups of naked axons; 3, large groups of naked axons; 4, confluent foci of demyelination; and 5, widespread demyelination. Hematoxylin-eosin staining was performed to grade the inflammation, by counting the number of perivascular mononuclear infiltrates and the number of cells per infiltrate. Brain sections were evaluated under (20×) magnification of optical fields [23]. Histological analyses were performed in the study groups consisting at least of 4 brains. Images were processed using Image-Pro Analyser version 7 for windows XP/Vista (Manufacture: Media Cybermetics, Inc. Rockville, MD, USA).

### 4.3. Statistical Analysis

The data are presented as mean ± SEM. Comparisons in the animal studies were made using a one way or repeated ANOVA, Tukey Post-hoc test, and, when mentioned, the unpaired *t*-test (normality assumed). For survival analysis, we used the Kaplan Meier survival curve. For cell culture and histological studies—the unpaired *t*-test was used (normality assumed). When normality was not assumed, the Mann Whitney U test was used. Statistical analysis was performed using IBM SPSS Statistics for Windows, version V.23 (IBM Corp., Armonk, NY, USA).

## 5. Conclusions

We show here the high efficacy of “CSF exchange therapy”, a novel therapeutic approach aimed at exchanging the endogenous pathogenic CSF with a new and healthy one, through drainage of the endogenous CSF and its continuous replacement with artificial CSF enriched with secretions of MSCs. Treatment of EAE-mice with CSF exchange therapy using the (in/out) enriched-aCSF protocol caused a significant delay in the onset and amelioration of the clinical paralytic symptoms, with a reduction in axonal damage and demyelination. A partial and smaller effect was found when using the treatment protocols of (in) enriched-aCSF (mostly delaying onset) and (in/out) aCSF (mostly ameliorating progression). The feasibility of CSF exchange therapy reported here in the EAE model might be applied to other neurodegenerative diseases of the CNS.

## Figures and Tables

**Figure 1 ijms-20-01793-f001:**
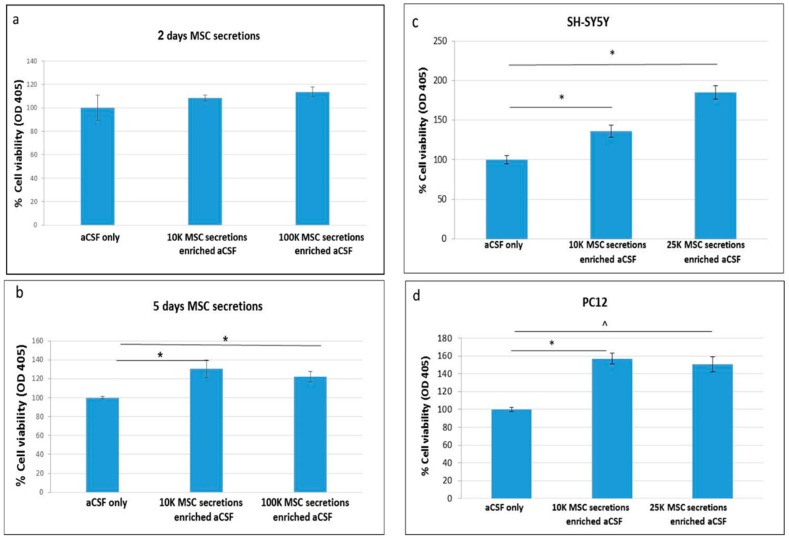
Artifical cerebrospinal fluid (aCSF) enriched with secretions of mesenchymal stem cells (MSCs) increases cell viability of PC12 and SH-SY5Y neuronal cell lines. (**a**,**b**) aCSF enriched with secretions of MSCs grown for 5 days, but not 2 days, increases cell viability of PC12 neuronal cell line: PC12 were treated ON with enriched-aCSF containing secretions of 10 or 100 K/mL MSCs, which were grown for 2 or 5 days in aCSF; (**a**) While secretions of 2 days growing 10 or 100 K/mL MSCs in aCSF did not show an increase in PC12 cell viability; (**b**) the “enriched-aCSF” containing the secretions of 5 days growing MSCs in aCSF did show a significant increase in the PC12 cell viability relative to unenriched-aCSF treated cells (*t*-test. * *p* = 0.017 and * *p* = 0.0065 for secretions of 10 and 100 K/mL MSCs, respectively, with a comparable effect of both cell concentrations); (**c**,**d**) Enriched-aCSF (secretions derived from 10–25 K/mL MSCs) increases cell viability of SH-SY5Y and PC12 neuronal cell lines: in both cell lines a significant increase in cell viability was found with the enriched-aCSF with secretions from 5 days grown MSCs 10–25 K/mL; (**a**) In SH-SY5Y: *t*-test, * *p* = 0.008 and * *p* = 0.00013 in 10 and 25 K/mL, respectively, relative to aCSF treated cells, with a stronger effect of the secretions of 25 K MSCs relative to 10 K/mL (* *p* = 0.0002); (**b**) In PC12: Mann Whitney U test, * *p* = 0.05 in 10 K/mL, with a similar trend in 25 K/mL, ^ *p* = 0.1, relative to aCSF treated cells. No significant difference in viability was detected between the secretions of 10 and 25 K/mL. OD_405_ = optical density at 405 nm.

**Figure 2 ijms-20-01793-f002:**
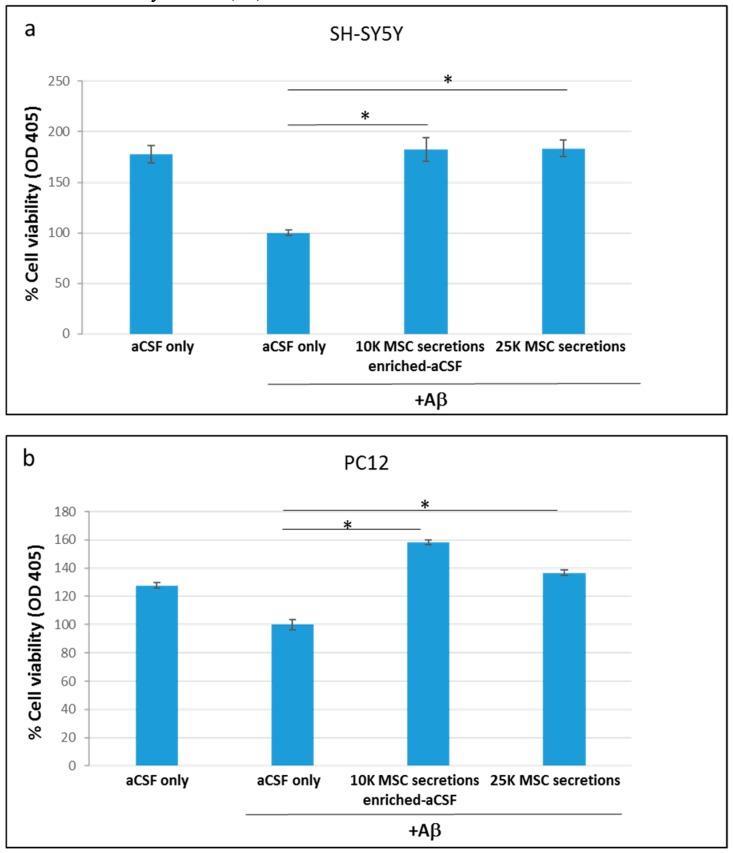
Enriched-aCSF increases cell viability of neuronal cell lines exposed to Aβ neurotoxin. (**a**) While SH-SY5Y cell viability was reduced under Aβ, a significant increase in cell viability was noticed in the enriched-aCSF (*t*-test, *p* = 0.0002, *p* = 0.00003, in 10 and 25 K/mL, respectively) with a comparable effect of the secretions of both MSCs concentrations; (**b**) Similarly, while PC12 cell viability was reduced under Aβ, a significant increase in cell viability was noted in the enriched-aCSF treated cells (10 K: increase of 58.1% (Mann Whitney U, *p* = 0.02), 25 K: increase of 36.5% (*t*-test, *p* = 0.00004) with a comparable effect of the secretions of both MSCs concentrations. OD_405_ = optical density at 405 nm.

**Figure 3 ijms-20-01793-f003:**
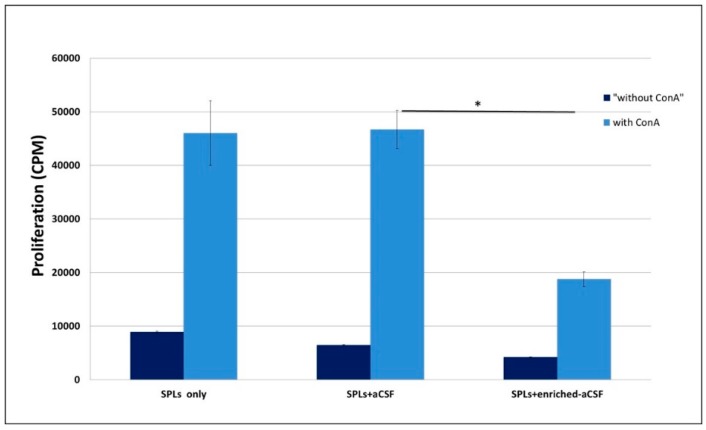
Suppression of splenocyte proliferation by the enriched-aCSF. Mouse splenocytes were grown with aCSF enriched with the MSC secretions or with unenriched-aCSF (or splenocytes only) and assayed for proliferation in response to ConA. The enriched-aCSF treated splenocytes showed significantly lower proliferation than the unenriched-aCSF treated mice (*t*-test, *p* < 0.0007). Unenriched-aCSF treated mice showed similar proliferation to splenocytes only (untreated). CPM = counts per minute. SPL = splenocytes.

**Figure 4 ijms-20-01793-f004:**
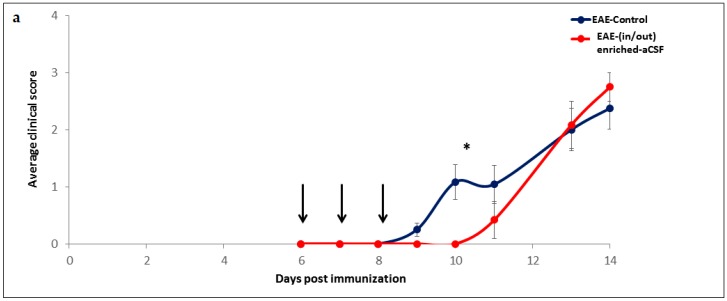
Short-term enriched-aCSF (in/out) exchange therapy shows a short-term amelioration of the EAE clinical symptoms. Experimental autoimmune (EAE)-mice treated with in/out MSC secretions enriched-aCSF (enriched-aCSF) exchange therapy delivered at days 6–8 following EAE-induction showed a trend of lower average clinical EAE-score at days 9–11 post EAE-induction (repeated ANOVA (f(1,16) = 3.73, *p* = 0.07)). (**a**) with a significant difference in the average clinical score at day 10 between the treated and untreated mice (*t*-test, *p* = 0.012). While untreated mice developed the disease already at day 9, treated mice developed it only at day 11 following induction (arrows represent days of treatment); (**b**) A significantly lower average cumulative score at days 0–10 post EAE-induction (*t*-test, *p* = 0.017) was detected in the treated mice relative to the untreated EAE-control; (**c**) A significantly lower average maximal score at days 0–10 post EAE-induction, (*t*-test, *p* = 0.012) was detected in the treated mice. From days 12–13 (i.e., 4–5 days after the end of therapy) the graphs of the average score of treated and untreated mice merged, and no significant differences were found in the cumulative and maximal scores; (**d**) Using the Kaplan Meier analysis revealed significant less EAE disease-free mice among the treated mice relative to the untreated (Log-Rank Test: *p* = 0.028), with the median values for the 50% chances to stay free of disease symptoms was at day 11 for the treated and at day 8 for the untreated mice.

**Figure 5 ijms-20-01793-f005:**
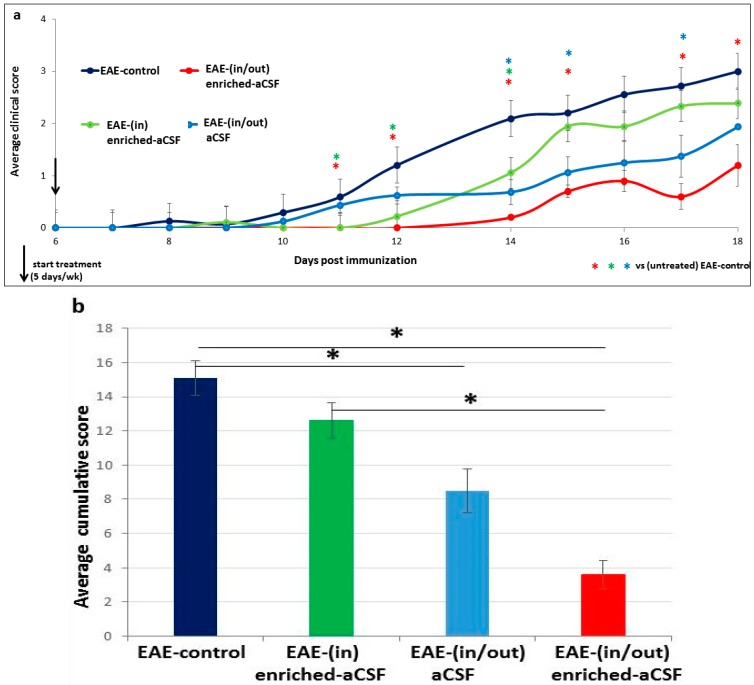
Prolonged amelioration of EAE clinical symptoms during prolonged CSF exchange therapy: (in/out) enriched-aCSF protocol was more effective than (in) enriched-aCSF and (in/out) aCSF. Prolonged CSF exchange therapy, which was delivered during days 6–22 after disease induction (performed 5 times a week for 2 weeks) was performed, comparing 3 protocols: (in/out) enriched-aCSF, (in) enriched-aCSF, and (in/out) aCSF (unenriched), vs. (untreated) EAE-control mice. Repeated ANOVA analysis showed a significant difference between the different treatment groups (f(2,24) = 9.59, *p* < 0.001)), with Tukey Post-hoc analysis showing significant differences between the control EAE-mice and each treatment group (*p* < 0.001, *p* = 0.035 and *p* = 0.001 for (in/out) enriched-aCSF, (in) enriched-aCSF, and (in/out) aCSF, respectively). (**a**) While the (in/out) enriched-aCSF mice showed a significant lower average score relative to control-EAE during all days of symptoms follow-up (*p* = 0.005, *p* = 0.013, *p* < 0.001, *p* = 0.004, *p* = 0.001, *p* = 0.005 for days 11, 12, 14, 15, 17, 18 post induction, respectively, red asterisks), the (in) enriched-aCSF mice showed a significant lower average score (*p* > 0.001, *p* = 0.01, *p* = 0.002 only at days 11, 12, 14, respectively, green asterisks; and the (in/out) aCSF mice showed a significant lower score (*p* < 0.001, *p* = 0.01, *p* = 0.019 only at days 14, 15, 17, with a similar trend, *p* = 0.066, also at day 18, respectively, light blue asterisks. A lower average score of (in/out) enriched-aCSF-mice was demonstrated relative to that of (in) enriched-aCSF-mice at days 15 and 17 (*p* = 0.06 and *p* = 0.03, respectively); (**b**) A significant difference in the average cumulative score between the groups was noticed (one-way ANOVA (f(3,44) = 11.04, *p* < 0.001)), with further analysis with Tukey Post-hoc showing that the (in/out) enriched-aCSF-mice and the (in/out) aCSF had lower scores compared to EAE-controls (*p* < 0.001, *p* = 0.008, respectively), with (in/out) enriched-aCSF-mice having lower scores than the (in) enriched-aCSF-mice (*p* = 0.004), with comparable scores of the (in) enriched-aCSF-mice and the (in/out) aCSF-mice; (**c**) A significant difference in average maximal score between the groups was noticed (one-way ANOVA (f(3,48) = 5.55, *p* = 0.003)), with further analysis with Tukey Post-hoc showing that the (in/out) enriched-aCSF-mice had a lower score compared to control and (in) enriched-aCSF (*p* = 0.002, *p* = 0.02, respectively), with comparable scores of the (in) enriched-aCSF-mice and the (in/out) aCSF-mice; (**d**) Kaplan Meier analysis revealed significant less EAE disease-free mice among the different groups (Log-Rank Test: *p* < 0.001), with each pair of groups showing significant differences ((in/out) enriched-aCSF-mice vs. EAE-control, (in) enriched-aCSF-mice and (in/out) aCSF-mice: *p* = 0.000387, 0.00507 and 0.0041, respectively; (in) enriched-aCSF-mice vs. EAE-control and (in/out) aCSF-mice: *p* = 0.0415 and 0.0141, respectively), but not (in/out) aCSF-mice vs EAE-control. The median values for the 50% chances to stay free of disease symptoms were 15, 13, 11, and 9 days for the (in/out) enriched-aCSF-mice, (in) enriched-aCSF-mice, (in/out) aCSF-mice and EAE-control, respectively (asterisks present comparison of a specific group with EAE-control).

**Figure 6 ijms-20-01793-f006:**
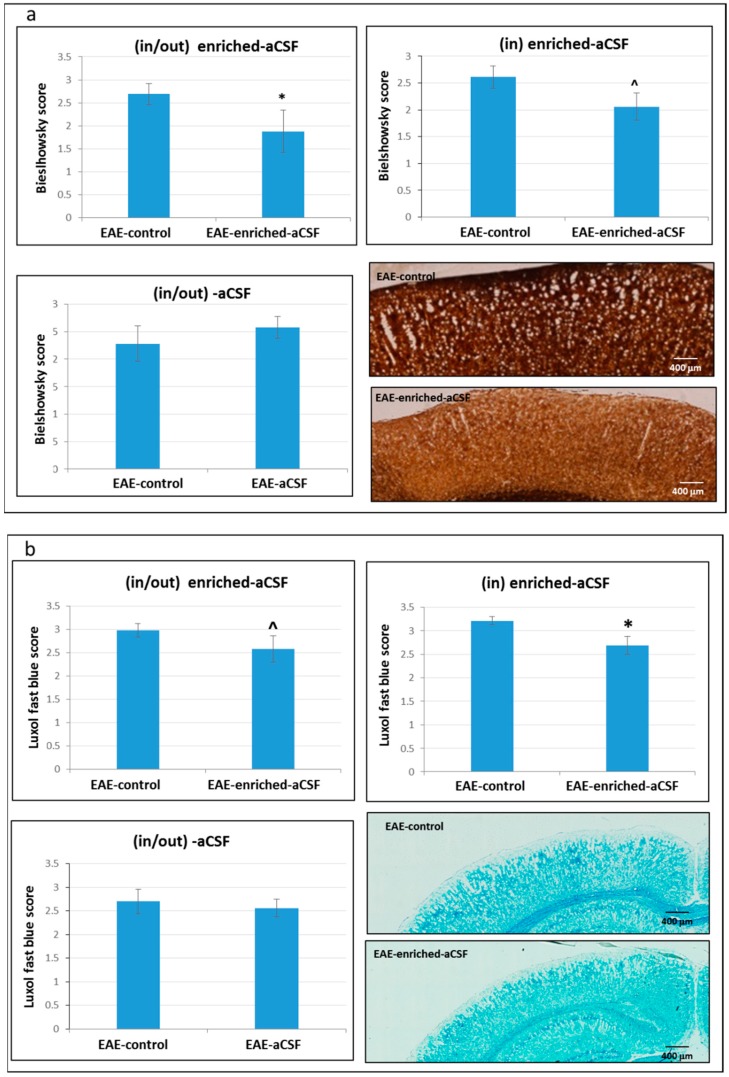
Reduced axonal damage and demyelination in the cortex of enriched-aCSF treated EAE-mice. (**a**) A significantly lower axonal damage in the Bielschowsky staining in the (in/out) enriched-aCSF treated mice relative to EAE-control mice was found (* *p* = 0.048). A similar trend was found in the prolonged treatment of (in) enriched-aCSF-mice relative to the EAE-control mice (^ *p* = 0.09), while no difference in the (in/out) aCSF-treatment vs EAE-control was detected; (**b**) A trend of less demyelination in the LFB staining in the (in/out) enriched-aCSF treated- mice relative to the EAE-control mice (^ *p* = 0.09), and a significant lower demyelination in the (in) enriched-aCSF-mice relative to the EAE-control mice (* *p* = 0.02) was found, and no difference in demyelination in the (in/out) aCSF-treatment vs EAE-control was shown.

**Table 1 ijms-20-01793-t001:** Factors secreted by MSCs growing in aCSF.

Factors Detected in the Enriched-aCSF (eCSF)		Concentration
BDNF		22.06 ± 4.83 pg/mL
CNTF		16.87 ± 12.50 pg/mL
TGF-β		14.13 ± 6.26 pg/mL
Anti-oxidant capacity	Enzymatic + non-enzymatic	0.46 ± 0.108 nmol/μL
	Non-enzymatic only	0.07 ± 0.012 nmol/μL

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
