# Peer review of "Cerebrospinal Fluid (CSF) Exchange with Artificial CSF Enriched with Mesenchymal Stem Cell Secretions Ameliorates Experimental Autoimmune Encephalomyelitis"

_ijms, 2019, doi:10.3390/ijms20071793_

Reviewer 1 Report

This study shows that a CSF exchange procedure reduced the severity of a multiple sclerosis model (MOG-induced EAE). In accordance with in vitro experiments showing that enriched-CSF increases cell viability of neuronal cell lines, the authors showed that the efficacy of this procedure is increase when the artificial CSF is enriched with mesenchymal stem cell secretions. This procedure induced a delay in disease onset with an amelioration of the clinical symptoms and a reduction in axonal damage and demyelination. 

Comments:

-in vitro experiments: precise the n of each group.

-The authors should include whether the data follow a normal distribution or not and apply normality tests. If there is no normality assumption, they should apply non-parametric test (like Mann Whitney U test). The t-test is correct only in case of normality assumption.

- There is no reference to the figure 3 in the text. 

- Figures 2, 3, 4 and 5: there is only the unit on the horizontal axis. Write more information on the horizontal axis (cell viability or index of proliferation for example).

 -Figure 6: From days 12 to 13 no significant differences in clinical score were noticed. What is the evolution of the clinical curve after 14 days? It seems that the evolution is less severe for the untreated group.

-Figure 6: Why the average cumulative score and the average maximal score are not performed on the same time? (average cumulative score: days 0-11; average maximal score: days 0-10).

- In this study the authors start the CSF exchange procedure before the EAE onset. Is the CSF exchange procedure efficient when apply after the first clinical symptoms? That may be clinically relevant.

- Bielschowsky and LFB staining: Indicate the region of the central nervous system on which the quantification is realized.

-Figure 8 shows a reduced axonal damage and demyelinisation in enriched-aCSF treated-mice. In EAE, the damages are caused by immune cell infiltration. However, we know that the CSF is a key cerebral invasion route for immune cells. The authors should check if the beneficial effect of the CSF exchange could be due to a reduced immune cells invasion in the central nervous system.

- Is the CSF exchange necessary? What is the effect of the injection of neurotrophic factors in the CSF?

Author Response

Reply to reviewer #1:

We would like to thank very much the reviewer for the thorough review of this paper.

Here are our replies to the reviewer’s comments.

Comment:.- in vitro experiments: precise the n of each group.

Reply:

In each group at least 3-4 replicates. We added it in the Methods

Comment:  -The authors should include whether the data follow a normal distribution or not and apply normality tests. If there is no normality assumption, they should apply non-parametric test (like Mann Whitney U test). The t-test is correct only in case of normality assumption.

Reply:

As requested, we have tested whether the data follow a normal distribution. In the cases of normality assumption we used the t-test, while where there was no normality assumption we used the Mann Whitney U test.  We have added this in the Methods and Results accordingly. The protection against H2O2 did not reach a statistical significance in the Mann Whitney U test, and it is presented now in the Results text. We have added to the results of the neuroprotection against amyloid in PC12 also results we have in SH-SY5Y cells (Fig 3 and text), showing significant neuroprotection by enriched-aCSF.

Comment: - There is no reference to the figure 3 in the text.

Reply:

This part was by mistake missing, and we have added it now. (Due to the request of the other reviewer to move fig 1 to supplement, fig 3 is now numbered as fig 2). 

Comment: - Figures 2, 3, 4 and 5: there is only the unit on the horizontal axis. Write more information on the horizontal axis (cell viability or index of proliferation for example).

Reply:

We have added this information.

Comment:   -Figure 6: From days 12 to 13 no significant differences in clinical score were noticed. What is the evolution of the clinical curve after 14 days? It seems that the evolution is less severe for the untreated group.

Reply:

There was no statistical difference at day 14 between the treated and untreated (p=0.5). We have added it in the results. [Latter, the animals were sacrificed for histological evaluation].

Comment:.-Figure 6: Why the average cumulative score and the average maximal score are not performed on the same time? (average cumulative score: days 0-11; average maximal score: days 0-10).

Reply:

We have changed now the fig of cumulative also to 1-10. We explained in the text that the cumulative score showed a trend of being lower in the treated mice also at days 0-11 (Mann Whitney U test p=0.06).

Comment:   - In this study the authors start the CSF exchange procedure before the EAE onset. Is the CSF exchange procedure efficient when apply after the first clinical symptoms? That may be clinically relevant.

Reply:

This is a very interesting issue and further studies will allow addressing it.  We added in the Discussion that additional studies are needed to provide further support to our findings, including studies to test the efficacy of the treatment once applied after the development of first clinical symptoms.

Comment: - Bielschowsky and LFB staining: Indicate the region of the central nervous system on which the quantification is realized.

Reply:

The quantification of Bielschowsky and LFB staining was performed in cortical regions. We added it in the Results.

Comment: -Figure 8 shows a reduced axonal damage and demyelinisation in enriched-aCSF treated-mice. In EAE, the damages are caused by immune cell infiltration. However, we know that the CSF is a key cerebral invasion route for immune cells. The authors should check if the beneficial effect of the CSF exchange could be due to a reduced immune cells invasion in the central nervous system.

Reply:

We indeed checked this option however no significant reduction of the peripheral inflammation (cell infiltrations) was observed as a direct result of the CSF exchange (as presented in in the last paragraph of the Results). We explained in the Discussion: ”We did not detect a decrease in infiltrate burden in the treated mice; such an effect can possibly be detected if tested at earlier stages, when the immune activation cascade starts”.

Comment: - Is the CSF exchange necessary? What is the effect of the injection of neurotrophic factors in the CSF?

Reply:

It seems from the results that exchange of CSF [both elimination of endogenous and infusion with enriched aCSF) is better than infusion only (termed here ”in” )]. We believe that injection in the CSF of neurotrophic factors may not be as efficient as injection of many beneficial factors including not only neurotrophic factors but also anti-inflammatory (which were detected here in the enriched aCSF, as were also anti-oxidants), as well as other compounds secreted by MSC (pool of many beneficial factors). In any case, our strategy speaks not only on a wide pool of components but also about continuous infusion (with continuous elimination), which has not been reported before, and maybe valuable.

Reviewer 2 Report

Valitsky et al demonstrate the therapeutic potential of CSF exchange in treating experimental autoimmune encephalomyelitis.  The experiments generate new and important information and the results are potentially important and of general interest.  However, there are several minor issues that need to be resolved before considering the manuscript for publication. 

-Adding a reference at the end of first paragraph of introduction.

- A comment in the intro or discussion on using human secretions for alleviating symptoms of an animal model.

-Figure 1 is poorly presented. The images of plates and syringe pumps aren't pretty and could be replaced by a written description. I suggest that the authors to have a look for a free web tool that can help in generating professional figures called BioRender (https://biorender.com/). The updated figure should be then moved to be a supplementary.

 - A comment on why the authors have selected two cancerous cell lines PC12 and SH-SY5Y and haven't considered trying to asses the cell viability on at least one primary cell line; preferably from human or even a rat/mice origin where the results will be more valuable for the desired aim of the study.

-Figures for cell viability need to be presented as percentages compared to what's considered as 100% viability (the control of each experiment). The actual new percentage values can also be added on the top of each bar to enhance visibility.

-X and Y axes need to be fully spelled. Adding any abbreviations could be done in brackets. The (a), (b) and (c) in figure legends are better to be added at the beginnings of sentences rather than the end. Indicating which statistical test has been used in the legend of each graph will enhance the readability.

-Comment on the results and/or discussion on why PC12 doesn't respond to 25K in the same way of SH-SY5Y in Figure 3; especially as it looks that the same trend rolls over to Figure 4 as well.

-Define CPM in Figure 5 Y-axis.

-Line 309 of page 13; explain Bielschowsky staining or provide a reference.

-Images in Figure 8a aren't clear. Scale bars should be in white to provide the needed contrast.
-Line 394 of page 15; you mentioned "other studies"but you only provide one reference.

Author Response

Reply to reviewer #2:

We would like to thank very much the reviewer for the thorough review of this paper.

Here are our replies to the reviewer’s comments.

Comment: -Adding a reference at the end of first paragraph of introduction.

Reply:

We have added such a reference.

Comment:.- A comment in the intro or discussion on using human secretions for alleviating symptoms of an animal model.

Reply:

We added in the Introduction: We use here secretions of MSCs of human origin, having the advantage of exploring its potential for future therapeutic purposes. These cells are suitable for use in animals since they present veto-like properties (Potian et al.) and suppress host rejection (Coulson-Thomas et al.).

Comment: -Figure 1 is poorly presented. The images of plates and syringe pumps aren't pretty and could be replaced by a written description. I suggest that the authors to have a look for a free web tool that can help in generating professional figures called BioRender (https://biorender.com/). The updated figure should be then moved to be a supplementary.

Reply:

We have changed the figure, using the BioRender, and moved it to be a supplementary.

Comment:  - A comment on why the authors have selected two cancerous cell lines PC12 and SH-SY5Y and haven't considered trying to asses the cell viability on at least one primary cell line; preferably from human or even a rat/mice origin where the results will be more valuable for the desired aim of the study.

Reply:

We have used these 2 commonly used neuronal cell lines to show protection in cells. We indeed agree that using primary cell line will be valuable. We have added in the Discussion that further studies using primary cells will be contributing.

Comment:  -Figures for cell viability need to be presented as percentages compared to what's considered as 100% viability (the control of each experiment). The actual new percentage values can also be added on the top of each bar to enhance visibility.

Reply:

We have changed now the figures of cell viability to be presented as %.

Comment:  -X and Y axes need to be fully spelled. Adding any abbreviations could be done in brackets. The (a), (b) and (c) in figure legends are better to be added at the beginnings of sentences rather than the end. Indicating which statistical test has been used in the legend of each graph will enhance the readability.

Reply:

We now fully spelled the axes. We have moved the a), (b) and (c) in figure legends to the beginnings of sentences. We now indicated which statistical test has been used in the legend of each graph

Comment:  -Comment on the results and/or discussion on why PC12 doesn't respond to 25K in the same way of SH-SY5Y in Figure 3; especially as it looks that the same trend rolls over to Figure 4 as well.

Reply:

The difference in response to 10K and 25k between the SH-SY5Yand PC12 in Figure 3, may be attributed to cell type differences. Yet, as both are cancerous cell lines, it will be contributing to test also primary neuronal cells, as we added in Discussion (as mentioned in comment 4 in response to the reviewer's comment). (Due to the reviewer's request to move fig 1 to supplement, fig 3 is now numbered as fig 2). 

Comment:  -Define CPM in Figure 5 Y-axis.

Reply:

We have defined in legend the CPM= counts per minute.

Comment:  -Line 309 of page 13; explain Bielschowsky staining or provide a reference.

Reply:

We changed in the Reults to: We performed Bielschowsky silver impregnation for axonal pathology. A reference is included in the Methods.

Comment:  -Images in Figure 8a aren't clear. Scale bars should be in white to provide the needed contrast.

Reply:

We have changed the color of the bars into white.

Comment:  -Line 394 of page 15; you mentioned "other studies" but you only provide one reference.

Reply:

We have changed to: “another study … has shown”.